# The Immunomodulatory Effects of Apigenin and Quercetin on Cytokine Secretion by the Human Gingival Fibroblast Cell Line and Their Potential Link to Alzheimer’s Disease

**DOI:** 10.3390/ph18050628

**Published:** 2025-04-26

**Authors:** Anna Kurek-Górecka, Małgorzata Kłósek, Radosław Balwierz, Grażyna Pietsz, Zenon P. Czuba

**Affiliations:** 1Department of Microbiology and Immunology, Faculty of Medical Sciences, Medical University of Silesia in Katowice, Jordana 19, 41-808 Zabrze, Poland; mklosek@sum.edu.pl (M.K.); gpietsz@sum.edu.pl (G.P.); zczuba@sum.edu.pl (Z.P.C.); 2Institute of Chemistry, University of Opole, Oleska 48, 45-052 Opole, Poland; radoslaw.balwierz@uni.opole.pl

**Keywords:** neurodegenerative disease, cytokines, apigenin, quercetin, periodontitis

## Abstract

**Background:** The link between periodontal pathogens, inflammation, and neurodegenerative processes, including Alzheimer’s disease (AD), is evident. *Porphyromonas gingivalis* and *Treponema denticola* release lipopolysaccharide (LPS), constituting a virulence factor that takes part in the brain inflammatory process. Human gingival fibroblasts (HGF-1) are a source of pro-inflammatory cytokines released during periodontal diseases. Propolis is a rich source of quercetin and apigenin, which exhibit anti-inflammatory and immunomodulatory activities, influencing the concentration of pro-inflammatory cytokines. Considering this aspect, models with stimulated HGF-1, followed by LPS and/or interferon-α (IFN-α), were used. **Aim:** This study was designed to evaluate the concentrations of selected cytokines produced by HGF-1, which may influence brain inflammation. The immunomodulatory effects of apigenin and quercetin were investigated by measuring the concentration of interleukin-1β (IL-1β), interleukin-6 (IL-6), interleukin-8 (IL-8), interleukin-15 (IL-15), and tumour necrosis factor (TNF-α). This study’s novelty is based on insights into the immunomodulatory effects of selected flavonoids by correlating the secretion of pro-inflammatory cytokines by gingival fibroblasts during periodontal disease with inflammatory processes in the brain. The cytotoxicity of apigenin and quercetin was estimated using the MTT assay. Fibroblasts were stimulated with LPS at 200 ng/mL and/or IFN-α at 100 U/mL concentration, followed by incubation with apigenin (25–50 µg/mL) and quercetin (25–50 µg/mL). Cytokine concentrations were measured using the xMAP technology. **Results:** The most pronounced and statistically significant reduction in cytokine levels, particularly IL-6 and IL-15, was observed for quercetin in both concentrations (25 µg/mL and 50 µg/mL), especially following LPS stimulation. Apigenin in both analysed concentrations also significantly decreased the level of IL-6. These results suggest that quercetin and apigenin may indirectly act as potential immunomodulators in preventing brain inflammation by inhibiting the inflammatory process in periodontitis; however, this should be confirmed in further studies.

## 1. Introduction

Flavonoids are plant-derived compounds that exhibit multidirectional activities, including antioxidant, anti-inflammatory, anticancer, immunomodulatory, anti-atherosclerotic, and anti-aggregation action [1,2,3].

Across the wide spectrum of flavonoid bioactivity, the anti-inflammatory and immunomodulatory properties, in particular, seem to be important in reducing inflammatory pathways [4,5].

The basic flavonoid structure is a C6-C3-C6 unit composed of 15 carbon atoms. A benzoyl ring and a phenylpropane unit form the structure of flavonoids (Figure 1A). The flavonoids are divided into groups based on the presence of a carbonyl group at the fourth carbon atom of the C-ring, the double bond between the second and the third carbon atoms of the C ring, and the number of hydrocarbon groups or other groups in the C ring.

The main dietary sources of flavonoids are vegetables like onions, tomatoes, pimentos, and broccoli; fruits, especially citrus fruits (lemon, orange, pomelo) and blueberries, as well as raisins and cranberries; and herbs and spices like cloves, cinnamon, nutmeg, ginger, and liquorice. In addition to these, a rich source of polyphenols is propolis [5,6]. Among the flavonoids found in Polish propolis are flavones (chrysin, apigenin, acacetin), flavonols (galangin, kaempferol, quercetin), and flavanones (pinostrobin and pinocembrin) [7].

Apigenin (Figure 1B) and quercetin (Figure 1C) show significant anti-inflammatory effects. Their mechanism of anti-inflammatory action is associated with the inhibition of 5-lipooxygenase (LOX-5) and cyclooxygenase-2 (COX-2). Therefore, they reduce the synthesis of prostaglandin E2 (PGE2), leukotrien B4 (LTB4), and thromboxane A2 (TxA2) [8,9,10,11]. This results in leucocytes being inhibited. Literature data indicate that apigenin and quercetin may prevent the development of neurodegenerative diseases [9,12,13].

In the course of Alzheimer’s disease (AD), accumulation of β-amyloid deposits is observed. The deposits are formed due to abnormal cutting of amyloid-beta precursor protein (APP protein) by β-secretase. As a consequence of this process, fibrillar tangles are formed. Quercetin has the ability to disaggregate β-amyloid fibrils [12]. According to Zhao Le et al., apigenin affects APP processing, prevents amyloid-β peptides (Aβ) accumulation, and reduces insoluble Aβ levels [14].

In addition to the presence of β-amyloid plaques and neurofibrillary tangles in the pathogenesis of AD, there is increasing evidence that inflammation, oxidative stress and lack of cholinergic transmission play a crucial role in the pathogenesis of AD. Moreover, chronic inflammation and uncontrolled immune responses are closely associated with AD.

The inflammatory pathways in the oral cavity are similar to those observed in the brain. Therefore, the link between oral pathogens, inflammatory pathways, and AD is evident [15,16,17,18]. It should be emphasised that oral bacterial infection causes the production of pro-inflammatory cytokines released by the host. The increased level of cytokines may lead to brain inflammation. *Porphyromonas gingivalis* and *Treponema denticola,* which are included in periodontal pathogens, release lipopolysaccharide (LPS), constituting a virulence factor [5,12].

Therefore, taking into consideration this aspect, we decided to use the model with stimulation HGF-1 followed by LPS and/or interferon-α (IFN-α). Based on our previous study, it is known that HGF-1 releases cytokines, followed by LPS and/or IFN-α stimulation [19]. In this context, considering the correlation between oral health and inflammation as well as systematic disease, including AD, the present study is designed to estimate the impact of quercetin and apigenin on selected pro-inflammatory cytokines such as interleukin-1β (IL-1β), interleukin-6 (IL-6), interleukin-8 (IL-8), interleukin-15 (IL-15), and tumour necrosis factor (TNF-α) secreted by gingival fibroblast during bacterial and viral infections. The novelty of this study is based on insights into the immunomodulatory effects of apigenin and quercetin. This is due to the correlation between the secretion of pro-inflammatory cytokines by gingival fibroblasts during periodontal disease and inflammatory processes in the brain.

## 2. Results

This study evaluated the cytotoxicity and immunomodulatory action of apigenin and quercetin.

### 2.1. The Impact of Apigenin and Quercetin on the Viability of HGF-1

The cell viability of HGF-1 cells after the application of apigenin and quercetin was determined using the MTT test. Apigenin and quercetin were used in the concentration ranges of 10, 25, 50, and 100 µg/mL. The results are depicted in Figure 2 and Figure 3 as well as in Appendix A. The highest cytotoxic effect was observed for apigenin at a concentration of 100 µg/mL. Apigenin at concentrations of 10, 25, 50, and 100 µg/mL decreases cell viability from 109.87% to 96.42%, 87.45%, 73.42%, and 60.64%, respectively. In the case of quercetin treatment, the cell viability of HGF-1 decreased from 94.76% to 91.71%, 71.87%, 62.42%, and 57.07%, respectively. Quercetin, similar to apigenin, exhibited the highest cytotoxic effect in regard to HGF-1 at a concentration of 100 µg/mL. For apigenin treatment, the differences were statistically significant, except for apigenin at a concentration of 10 µg/mL, followed by LPS stimulation. For the effect of quercetin treatment on HGF-1 cell viability in the experimental models used, the differences were statistically significant, except for quercetin at 25 µg/mL followed by LPS stimulation (Figure 3). As the determination of cytotoxicity by the MTT assay indicates limited cellular metabolism rather than direct cell death, apigenin and quercetin at concentrations of 25 and 50 µg/mL were used for further studies.

### 2.2. The Impact of Apigenin and Quercetin on Pro-Inflammatory Cytokines Secretion by HGF-1 Followed by LPS and/or IFN-α Stimulation

The effect of apigenin and quercetin on the production of selected cytokines such as IL-1β, IL-8, IL-6, IL-15, and TNF-α was estimated via an experimental model involving apigenin in HGF-1, followed by LPS and/or IFN-α stimulation. The results are presented in Figure 4 and in Appendix A.

Based on our previous study [19], it was confirmed that LPS and/or IFN-α induced an increased production of examined pro-inflammatory cytokines in order to increase cytokine concentration in the control cell line.

Some differences were noted when examining the effects of apigenin and quercetin at concentrations of 25 and 50 µg/mL on the cytokine profile in HGF-1, followed by LPS and/or IFN-α stimulation. The statistical significance variance of IL-1β was observed for quercetin at both concentrations of 25 and 50 µg/mL, followed by LPS stimulation, with *p*= 0.001 and *p* = 0.002, respectively. Notably, stimulation with LPS did not cause an increase in the level of IL-β in a cell line, but there was an observed statistically significant increase in the level of IL-1β for quercetin in both concentrations, followed by LPS stimulation (Figure 4a). It is, therefore, difficult to describe the influence of quercetin on this inflammatory marker in a consistent way in the case of IL-1β.

In case of IL-6, the statistical significance of the variance was noticed for quercetin in both concentrations of 25 and 50 µg/mL in three models of stimulation (LPS and/or IFN-α) (Figure 4d). A similar situation was observed for apigenin at a concentration of 25 µg/mL. However, apigenin at a concentration of 50 µg/mL exhibited significant statistical variation in the case of LPS (Figure 4c). Quercetin caused a significant statistical decrease in the level of IL-6 in both concentrations of 25 and 50 µg/mL, followed by LPS stimulation, with *p* = 0.004 and *p* < 0.001, respectively. Also, when quercetin was administered at a concentration of 50 µg/mL, followed by IFN-α stimulation, a significant statistical decrease was observed (*p* < 0.001).

In order to observe IL-8 behaviour, a significant variation was observed for apigenin at a concentration of 50 µg/mL, followed by LPS stimulation (Figure 4e). Apigenin caused a substantial decrease in IL-8 level (*p* = 0.049). For quercetin, significant statistical differences were found, followed by IFN-α and LPS+IFN-α stimulation in both concentrations (25 and 50 µg/mL) (Figure 4f). Quercetin, at its lowest concentration in the two models mentioned, led to a significant statistical increase in the concentration of IL-8, respectively, for the IFN-α and LPS+IFN-α models (*p* = 0.000 and *p* = 0.000).

Regarding the concentration of IL-15, it was noticed that only quercetin in both used concentrations caused statistically significant variation in all models of HGF-1 stimulation (followed by LPS, IFN-α, and LPS + IFN-α). Quercetin significantly reduced the level of IL-15 (Figure 4h). All outcomes achieved statistically significant variation, respectively, in the model with LPS for concentrations of 25 and 50 µg/mL (*p* < 0.001, *p* = 0.005). Regarding stimulation with IFN-α, quercetin exhibited a significant reduction in IL-15 in both concentrations, with *p* = 0.000 and *p* = 0.002, respectively. Considering the model with LPS+IFN-α, the behaviour was similar for concentrations of quercetin 25 and 50 µg/mL, respectively (*p* < 0.001 and *p* = 0.000).

The last estimated cytokine was TNF-α. In this case, the statistically significant variation was noticed only for quercetin at a concentration of 50 µg/mL, followed by IFN-α stimulation (Figure 4j). Quercetin statistically significantly increased the level of TNF-α (*p* = 0.011). However, we noticed that IFN-α did not cause an increase in TNF-α; therefore, similarly to IL-1β, it is difficult to evaluate the impact of quercetin on this inflammatory marker, followed by IFN-α stimulation. The mechanism is unknown.

### 2.3. Statistical Analysis Comparing Effects of Apigenin and Quercetin on the Production of Selected Cytokines in LPS, IFN-α and LPS+IFN-α Stimulated HGF-1 Cells

The analysis of the results shows that, independent of concentration, quercetin exerts a stimulating effect on the production of IL-1β, followed by LPS stimulation. Both apigenin and quercetin demonstrate an inhibitory effect on the production of IL-6, especially in sequences where HGF-1 cells are stimulated, followed by LPS. Conversely, we observed that apigenin at a concentration of 50 µg/mL exhibits an inhibitory effect on the production of IL-8. Thus, quercetin exerts a stimulatory effect on the production of IL-8 in both concentrations, followed by IFN-α and LPS+IFN-α stimulation. In the case of quercetin, its inhibitory effect is mainly limited to IL-15 (Figure 4h). However, in the case of TNF-α, quercetin exerts a stimulating effect at a concentration of 50 µg/mL, followed by IFN-α stimulation. Given these observations, we re-analysed all data using more advanced statistical methods.

Hierarchical clustering analysis (HCA) and principal component analysis (PCA) are the most widely used tools for exploring similarities and hidden patterns between samples. In order to analyse the effects of apigenin and quercetin on the levels of pro-inflammatory cytokines evaluated in HGF-1 cells stimulated with LPS, IFN-α and LPS + IFN-α, the above-mentioned analyses were used. The results of the HCA analysis are depicted in Figure 5 and Figure 6, while the results of the PCA analysis are depicted in Figure 7.

The HCA was established on the Euclidean distance defined between sets. The resulting dendrogram of cytokines showed that all data could be clustered into three main groups (Figure 5). The first cluster group demonstrates similarities in the behaviour of IL-1β and TNF-α concentrations. The depicted effects are illustrated in Figure 5 and Figure 6, showing a similarity in the effects of apigenin and quercetin on these cytokines. Nevertheless, quercetin exhibited statistical significance, stimulating the effect obtained for IL-1β, followed by LPS (Figure 6). Additionally, quercetin at a concentration of 50 µg/mL, followed by IFN-α stimulation, caused a statistically significant increase in TNF-α. In the second cluster, IL-6 and IL-8 demonstrate comparable behaviour, but in the case of IL-8, quercetin demonstrates a stimulating effect, resulting in the increase in concentration of IL-8, followed by IFN-α and LPS+IFN-α stimulation (Figure 6). The third cluster demonstrates that the behaviour of IL-15 was totally different in comparison to all estimated cytokines. In the case of IL-15, quercetin was statistically significant in reducing the concentration of the mentioned cytokine (Figure 6).

The hierarchical clustering analysis obtained for the tested samples of apigenin and quercetin (Figure 6) indicates that similar results were obtained for the stimulation of HGF-1 by IFN-α and LPS + IFN-α in the case of quercetin. This observation is consistent with the results of the ANOVA test performed for IL-8.

In addition, it is noteworthy that quercetin exhibits a more uniform effect by forming a compact cluster structure, which we can see in Figure 6 at the bottom. In contrast to quercetin, apigenin forms multiple subclusters. The analysis appears to show differential action of both compounds.

A graph of the results of the PCA was generated, including the average effect of the different concentrations of apigenin and quercetin on the selected cytokines (Figure 7). The results of the PCA analysis validate the observations of the HCA analysis. The analysis confirms the clear differences in the performance of quercetin and apigenin. Apigenin and quercetin occupy a variety of surfaces. Confirming this observation, apigenin and quercetin form separate clusters in HCA. Based on PCA, it is noticeable that quercetin is found close to IL-1β and TNF-α, indicating a stronger effect of this compound on these cytokines. Moreover, the PCA reveals data indicating that both apigenin and quercetin exhibit differentiated action depending on the tested dose.

## 3. Discussion

New comprehensive oral health studies demonstrate the link between oral pathogens, inflammation, and AD [5,18,20,21].

Pro-inflammatory cytokines are produced by the host in response to oral bacterial infection. This leads to increased inflammation and may contribute to the encephalitis that occurs in people with Alzheimer’s disease. It is hypothesised that the immunomodulatory properties of natural products and their compounds may influence the concentration of pro-inflammatory cytokines and prevent brain inflammation. Substances exert non-specific effects by either increasing or inhibiting the release of inflammatory mediators, thereby stimulating both innate and adaptive mechanisms. Bee products, such as propolis, exhibit potent immunomodulatory properties and may also have positive effects on cognitive impairment associated with AD. Moreover, flavonoids present in propolis, like apigenin and quercetin, demonstrate anti-inflammatory and immunomodulatory activity, but also potentially prevent the accumulation of β-amyloid deposits. Our previous study showed that propolis and its component caffeic acid phenethyl ester influence the levels of pro-inflammatory cytokines released by HGF-1, followed by LPS, IFN-α, as well as LPS+IFN-α stimulation [19]. We decided to investigate the effect of other propolis components on the profile of selected pro-inflammatory cytokines released by gingival fibroblasts, based on the results of previous studies. According to Bachiega et al. [22], the immunomodulatory action of propolis is connected with the increased release of IL-1β.

However, the immunomodulatory action of propolis is a result of a synergistic effect of its components. Considering the effect of selected flavonoids like apigenin and quercetin with proven anti-inflammatory and immunomodulatory effects, we noted that stimulation followed by LPS and/or IFN-α did not induce the production of IL-β. It is a pro-inflammatory cytokine which participates in periodontitis. Thus, chronic inflammation during periodontitis is linked to the pathogenesis of AD [23].

IL-1β is mainly secreted by macrophages and dendritic cells. However, not only the cells of the immune system but also other cells like gingival fibroblasts, periodontal ligament cells, and osteoblasts also release IL-1β [24]. However, the secretion of IL-1β in the experimental model by the gingival fibroblast was not observed. However, in another study conducted in J774.2 macrophages, researchers demonstrated that apigenin and kemferol inhibited IL-1β gene expression [25].

IL-1β at the site of inflammation is responsible for increased local blood flow, leukocyte recruitment and neutrophil infiltration. IL-1β promotes the secretion of matrix metalloproteinases (MMPs), receptor activators for nuclear factor κ B ligand (RANKL), PGE2, IL-6, and IL-8. In this context, in response to IL-1β, the fibroblasts in the periodontium produce pro-inflammatory mediators. On the one hand, the strategy of blocking IL-1β is a new therapeutic perspective; on the other hand, the short-term stimulation of IL-1β release by natural products is associated with their immunomodulatory effects. In our study, we noticed that quercetin, in contrast to apigenin, caused a statistically significant increase in IL-1β at both concentrations (25 and 50 µg/mL), followed by LPS stimulation, although LPS did not induce the increase in this marker. Therefore, it is necessary to continue our study in this case. In our previous study, we observed that propolis and CAPE did not induce significant changes in the concentration of IL-1β [19].

The immunomodulatory activity of selected flavonoids may be correlated with a decrease in the level of IL-6. In addition, Machado showed that green propolis reduced the levels of IL-6 and TNF-α in pneumonia induced by LPS [26].

Also, Zamarrenho et al. demonstrated that propolis reduces the concentration of IL-6 [27].

We noticed that apigenin and quercetin at both analysed concentrations (25 and 50 µg/mL) in the model with LPS also reduced the IL-6 level (Figure 4c,d). In our previous study, CAPE also caused a significant reduction in IL-6 levels, followed by stimulation of HGF-1 by LPS [19]. Moreover, the inhibitory effect of quercetin on IL-6 production by LPS-stimulated neutrophils was confirmed by Liu et al. [28]. It should be emphasised that the mentioned studies were conducted in vitro, therefore the confirmation should be provided in vivo studies. However, the study in an animal model showed that apigenin decreased the concentration of TNF-α and IL-1β at 24 h, followed by LPS injection in the serum of Wistar rats [29]. The results obtained in the LPS model are important in the context of the pathogenesis of AD. Periodontitis is connected mainly with Gram-negative bacterial infections, and many studies underline the correlation between endotoxin and AD [30,31,32]. The LPS used in the research model reflects a bacterial infection in the oral cavity, therefore the results obtained in this model appear to be more relevant than the model with IFN-α, which reflects a viral infection.

IL-6 is considered one of the most sensitive indicators of an inflammatory response. IL-6 has been demonstrated to play a pivotal role in the defence of an organism in instances where tissue injury has been detected by the immune system. Increases in the levels of this cytokine are a consequence of an immune system response to aggression. However, it is important to note that persistent secretion of IL-6 can contribute to the maintenance of an inflammatory process and eventually lead to the development of chronic inflammatory disease. Therefore, chronic inflammation may be prevented by reducing IL-6 levels with the studied flavonoids. It has been hypothesised that the impact of the treatment with apigenin or quercetin may also be beneficial for patients with AD, given the increased levels of inflammatory mediators, including IL-6, observed in this group. It is evident that IL-6 plays a pivotal role in the pathogenesis of AD. The study conducted by Cojocaru et al. demonstrates that elevated levels of peripheral IL-6 secretion may be responsible for the acute-phase proteins observed in the serum of AD patients [33]. Scientists indicate that flavonoids such as apigenin, quercetin, and luteolin may be useful in the therapy of neuroinflammation due to their impact on pro-inflammatory mediators, including TNF-α, IL-6, IL-1β, and NO [33,34,35].

In vitro studies have confirmed that TNF-α, IL-1β, and IL-6 induce Aβ42 (beta-amyloid) synthesis and phosphorylation of P-Tau (abnormal P-Tau phosphorylation causes damage to neuronal cytoskeleton) [28]. Therefore, the inflammatory theory underlined that elevated level of IL-6, IL-1β, and TNF-α is associated with a greater incidence of intellectual ability loss in elderly patients. Considering this hypothesis and taking into account that periodontal disease can directly contribute to the peripheral inflammatory environment, the impact of flavonoids on TNF-α seems reasonable. TNF-α is the most common early signal molecule during inflammation. However, generally studied flavonoids did not have a statistically significant effect on changes in the concentration of this cytokine.

Another cytokine assayed, the increased expression of which may be a result of chronic parotitis, is IL-8. IL-8 is required for the migration and action of neutrophils involved in acute and chronic inflammation and chronic inflammation. Apigenin at the highest concentration of 50 µg/mL caused a reduction in IL-8, followed by LPS stimulation. Comparing this result to the survival of cells after apigenin was applied in higher concentrations, it should be noted that survival decreased by more than 40%, so it is possible that the effect on the reduction in pro-inflammatory IL-6 concentration would be even higher with 100% survival. However, it is difficult to speculate on the outcome given the limitations of the MTT test method. Determination of cytotoxicity by the MTT test indicates restricted cell metabolism and not direct cell death. Given this finding, further studies should consider determining cell survival at different time intervals. The MTT assay evaluates mitochondrial metabolic activity and does not allow for a clear distinction between viable cells and those undergoing apoptosis or metabolic slowdown. According to the ISO 10993-5 standard [36] for biological evaluation of medical devices, cell viability below 70% is considered indicative of cytotoxicity [37]. In the present study, flavonoid concentrations of 25 and 50 µg/mL did not reduce cell viability below this threshold (with the exception of quercetin at 50 µg/mL), and can therefore be regarded as non-cytotoxic. It is also worth noting that modulation of cellular metabolic activity by bioactive compounds, including flavonoids, is often reported in the literature as a regulatory rather than destructive effect on cells [38]. Generally, apigenin is considered to be relatively less toxic among the flavones [39]. In our previous study, CAPE and propolis also elevated IL-8. The mobilisation of the immune system to defend itself may explain this.

In our study, we measured the concentration of pleiotropic cytokine–IL-15. It plays a key role in initiating and maintaining inflammation. Our observation clearly indicates that, in contrast to apigenin, quercetin caused a decrease in this cytokine in both concentrations. Referring to our earlier studies conducted on the same research model, Polish propolis showed similar behaviour towards this cytokine [19].

In light of these findings, the correlation between the inflammatory process in the oral cavity and the pathogenesis of AD is emphasised. Apigenin and quercetin are components of propolis that influence the concentration of pro-inflammatory cytokines, especially IL-6 and IL-15. It should be emphasised that apigenin exhibits a reduction in IL-8, followed by LPS stimulation, in contrast to quercetin. It is especially interesting, due to the fact that apigenin exhibited the immune-regulatory activity in vivo [40]. The observed impact of apigenin and quercetin may potentially influence inflammatory pathways in the brain, however, this fact should be confirmed in further studies. Our preliminary observations in vitro require further confirmation.

## 4. Materials and Methods

### 4.1. Materials

Quercetin and apigenin were delivered from Sigma-Aldrich (Darmstadt, Germany). Dimethyl sulfoxide (DMSO) was purchased from Sigma Chemical Company (St. Louis, MO, USA). Lipopolysaccharide BE. Coll 026:B6 was purchased from Difco Laboratories (Detroit, MI, USA); thus, Interferon-alpha (IFN-α) 3 MIU/0.5 mL was purchased from Schering-Plough, Brinny, Ireland. The human gingival fibroblast cell was delivered from the American Type Culture Collection (ATCC, Manassas, VA, USA). Dulbecco’s Modified Eagle’s Medium (DMEM), foetal bovine serum (FBS), and Trypsin-EDTA were delivered from Sigma Aldrich (Darmstadt, Germany). An antibiotic mixture (penicillin 10,000 U/mL/streptomycin 10,000 µg/mL) was delivered by Merck (Darmstadt, Germany).

### 4.2. HGF-1 Collection

The collection was conducted on the HGF-1. These cells were isolated from the gingival of healthy patients. The HGF-1 cells were grown at 37 °C with 5% CO_2_ in the incubator. Dulbecco’s Modified Eagle’s Medium (DMEM)-enriched L-glutamine was modified by ATTC. It contained 4.5 g/L glucose and 1.5 g/L sodium bicarbonate, and was supplemented with 10% foetal bovine serum (FBS). To ensure proper growth, a mixture containing 100 U/mL penicillin and 100 µg/mL streptomycin was added to the medium. To remove traces of serum, the cells were then detached using 0.25% (*w*/*v*) trypsin with 0.53 mM EDTA. Medium was added to the cells, and the cell suspension was centrifuged after 15 min. The sediment obtained was dissolved in the medium. In order to evaluate the number of cells, a Bürker counting chamber was used. The number of cells was counted according to Formula (1) [19].Number of cells in 1 mL = 4 squares counted 2 × 100 × 1000,(1)

Cell counts were diluted to 100,000 cells/mL. A quantity of 200 µL cultured cells was seeded into 96-well plates in the presence of LPS and/or IFN-α with or without apigenin or quercetin for 24 h. The stimulation by LPS at a concentration of 200 µg/mL and/or IFN-α at a concentration of 100 U/mL was led by 2 h. Then, in the experiment, HGF-1 was treated with apigenin and quercetin at concentrations of 25 and 50 µg/mL.

### 4.3. The Cytotoxicity Assay–MTT Assay

3-(4,5-dimethyl-2-thiazyl)-2,5-diphenyl-2H-tetrazolium (MTT) assay was used to determine cell viability. This is based on reducing 3-(4,5-dimethyl-2-thiazyl)-2,5-diphenyl-2H-tetrazolium bromide, which is transferred from viable cells to a blue formazan crystal. Apigenin and quercetin at the final concentrations of 10, 25, 50, and 100 µg/mL with or without LPS, and/or IFN-α, were added to 96 wells, such that each well contained 200 µL. Four controls were prepared for their experimental model: control of DMSO, control of LPS, control of IFN-α, as well as control of LPS+IFN-α. After 24 h, the medium was removed, and 180 µL of medium and 20 µL of MTT solution (5 mg/mL PBS) were added to each well. The cells were allowed to incubate for four hours. The supernatant was removed, and DMSO was added to each well to dissolve the resulting formazan crystals. The medium contains no other substances. The estimation was carried out as follows: spectrophotometrically at a wavelength of 550 nm.

The results, expressed in terms of absorbance, were calculated according to Formula (2) [41].% cell viability = sample absorbance 100/absorbance of the control,(2)

### 4.4. Multiplex Bead-Based Cytokine Assay

A multiplex assay was used for cytokine detection. The assay is based on fluorescent pigment-labelled antibodies. In the above-mentioned test, the ability of biotinylated antibodies to bind to cytokines by conjugation with streptavidin-phycoerythrin is tested. Cytokines were released from the HGF-1 cell line treated with 50 µL of apigenin and quercetin (25–50 µg/mL–final concentrations). Cytokines were measured in the culture supernatant. The supernatant was collected 24 h after incubation with the flavonoids tested. Standard dilutions and blanks were prepared using selected cytokine reference kits. In the conducted experiment, Bio-Plex Human cytokine Panel (BIO-RAD Laboratories, Hercules, CA, USA) was utilised in cooperation with Bio-Plex 3D Suspension Array System based on xMAP technology (BIO-RAD Laboratories Inc.). The supernatant derived from native HGF-1 (1 × 10^6^/mL), stimulated by LPS and/or IFN-α, after incubation with and without quercetin and apigenin at concentrations ranging from 25 to 50 µg/mL for 24 h, was analysed. Firstly, 50 µL beads were added to the wells. This was followed by a double wash with 100 µL buffer. ELx 50 magnetic washer (Winooski, VT, USA) was used to wash out the ferromagnetic beads. Then, 50 µL of standard dilution, samples and blank were incubated for 30 min on a shaker. After that, this was followed by a three-fold rinsing with 100 µL buffer and a 25 µL/well mixture of biotinylated antibodies was added. After 30 min incubation and washing, 50 µL of streptavidin-phycoerythrin conjugation was added and incubated. Then, three-fold rinsing with 100 µL of buffer was followed again. The beads were resuspended in 125 µL of buffer. The assay was conducted in a single well. The quantification process was conducted utilising the BioRad BioPlex 3D instrument, which employs a two-laser flow cytometry technique as its underlying methodology. One laser was used to detect the colour of the beads (assigned to a specific analyte), and a second was used to excite the phycoerythrin to determine the fluorescence (the measurement proportional to the amount of analyte bound). The concentrations of the analytes were read from the curves using appropriate standards. The experiments were repeated three times. The detection limits of the selected cytokines—IL-1β, IL-6, IL-8, IL-15, and TNF-α—expressed as the lowest standard concentrations, were 0.3, 0.36, 0.92, 19.49, and 3.81 pg/mL, respectively.

### 4.5. Statistical Analysis

All analyses were performed in triplicate, and the results were expressed as mean values. Statistical evaluations were conducted using STATISTICA 13.1 software (StatSoft Inc., Tulsa, OK, USA). The dataset underwent hierarchical cluster analysis (HCA) with the full linkage method and Euclidean distance, alongside principal component analysis (PCA). The PCA model was constructed using the iterative NIPALS algorithm, with a convergence threshold of 0.00001 and a maximum of 50 iterations. The number of components was determined based on optimal predictive capability through multiple cross-validation, with an upper limit set. The final PCA model was reduced to two principal components.

The PCA results, visualised in the loading plot of PC1 versus PC2, allowed for the identification of key variables influencing dataset variability and the determination of significant correlations among them. Both PCA and HCA were utilised to detect natural data clusters and assess differences in the effects of apigenin and quercetin on HGF-1 fibroblasts.

The impact of varying apigenin and quercetin concentrations on pro-inflammatory cytokine production, induced by LPS, IFN-α, and their combination, was analysed using one-way analysis of variance (ANOVA). Additionally, the MTT assay results were assessed via one-way ANOVA. Statistical significance was defined as *p* < 0.05.

## 5. Conclusions

Apigenin and quercetin can be considered natural modifiers of the immune response, affecting the inflammatory process. Both compounds, apigenin and quercetin, demonstrate an inhibitory effect on the production of IL-6, followed by LPS stimulation. Apigenin, in contrast to quercetin, at a concentration of 50 µg/mL, exhibits an inhibitory effect on the production of IL-8. Thus, quercetin exerts a stimulatory effect on the production of IL-8 at both concentrations, followed by IFN-α and LPS+IFN-α stimulation. These results are interesting and require further observation. In the case of quercetin, its inhibitory effect is mainly limited to IL-15. In this context, apigenin and quercetin represent potential ingredients that could be used in the prevention and treatment of AD. Further studies are needed to explain the strong impact of natural immune modifiers on the course of inflammation. However, due to the evidence of a link between the peripheral inflammatory processes and neurodegenerative diseases, the conducted studies seem to be relevant. Future research into the relationship between periodontitis and Alzheimer’s disease should focus primarily on the possibility of preventing the onset of periodontal disease. Our observations can provide a basis for future research directions.

## Figures and Tables

**Figure 1 pharmaceuticals-18-00628-f001:**
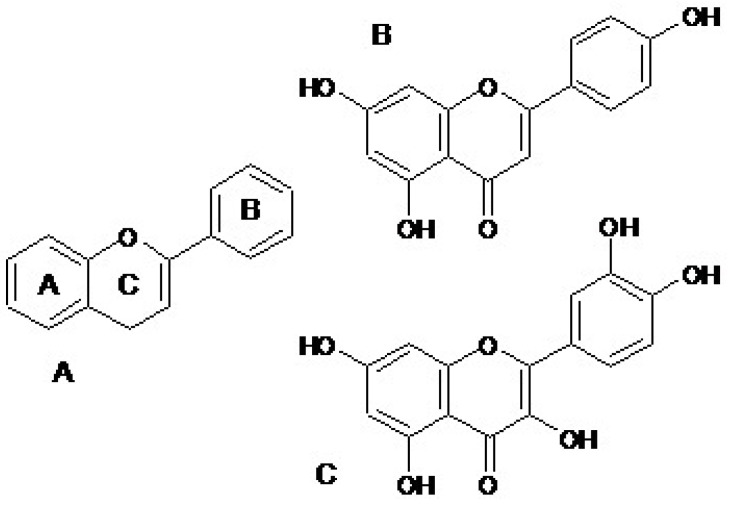
The basic chemical structure of flavonoids (**A**) and chemical structure of selected flavonoids like flavones-apigenin (**B**), flavonols-quercetin (**C**).

**Figure 2 pharmaceuticals-18-00628-f002:**
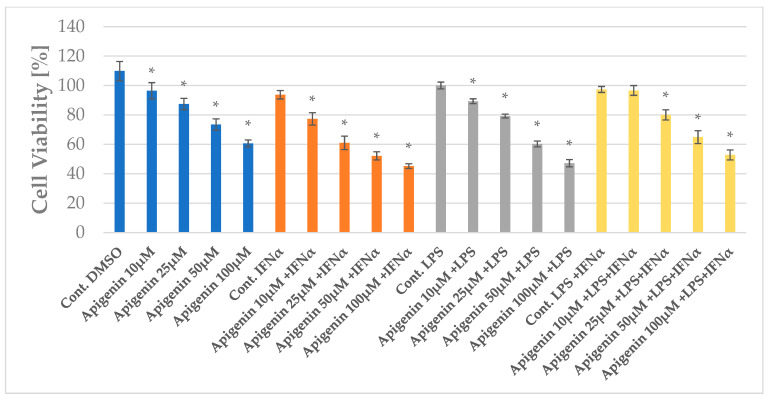
HGF-1 viability assayed by MTT (%)–the cytotoxic impact of apigenin. The values represent the mean ± SD of three independent assays. * mean *p* < 0.05 (calculated using LSD Test).

**Figure 3 pharmaceuticals-18-00628-f003:**
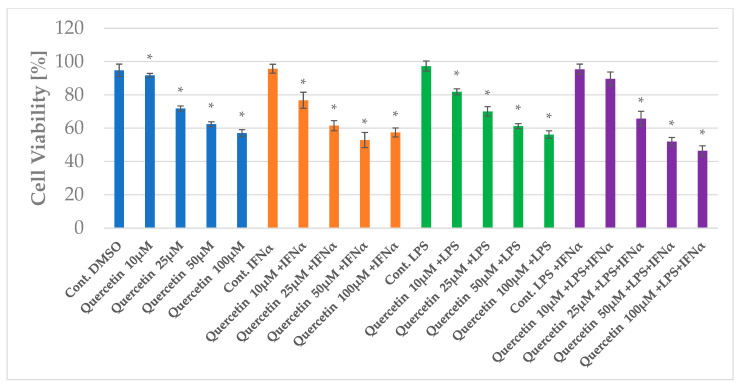
HGF-1 viability assayed by MTT (%)—the cytotoxic impact of quercetin. The values represent the mean ± SD of three independent assays. * mean *p* < 0.05 (calculated using LSD Test). Cont. DMSO—control of DMSO (native HGF-1, medium and 0.1% DMSO), Cont. IFN-α—control of IFN-α (native HGF-1, medium and IFN-α at concentration of 100 U/mL), Cont. LPS—control of LPS (native HGF-1, medium and LPS at concentration of 200 ng/mL), Cont. LPS+IFN-α—control of LPS combined IFN-α (native HGF-1, medium, LPS at concentration of 200 ng/mL, and IFN-α at concentration of 100 U/mL).

**Figure 4 pharmaceuticals-18-00628-f004:**
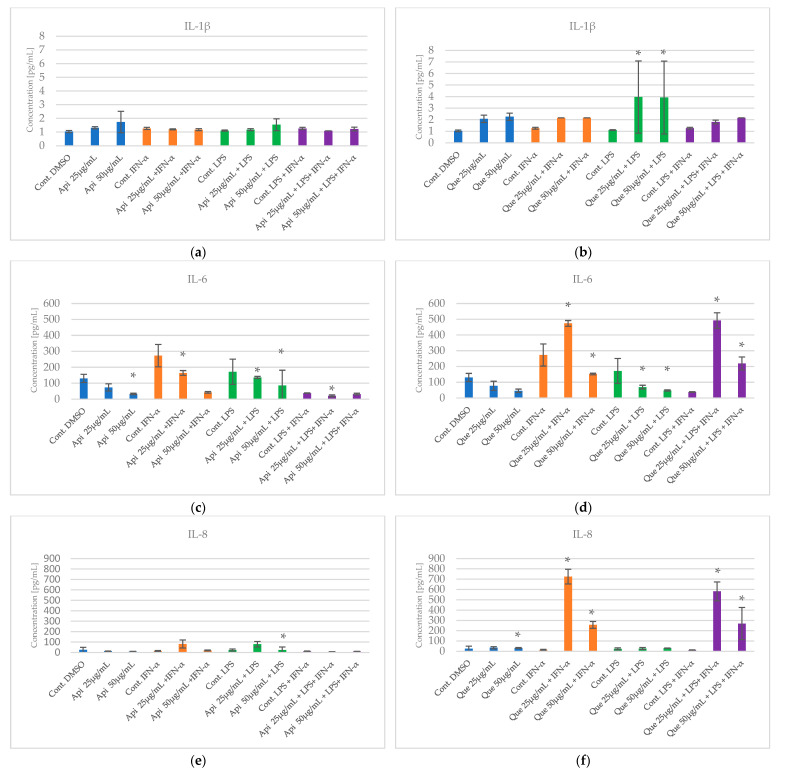
The impact of apigenin and quercetin on production of IL-1β (**a**,**b**), IL-6 (**c**,**d**), IL-8 (**e**,**f**), IL-15 (**g**,**h**), TNF-α, (**i**,**j**) in native and stimulated HGF-1 by LPS, IFN-α, and combination of LPS+ IFN-α. The presented values mean ±SD of three independent experiments (n = 8); * means *p* < 0.05 (calculated using LSD Test). Cont. DMSO—control of DMSO (native HGF-1, medium and 0,1% DMSO), Cont. IFN-α—control of IFN-α (native HGF-1, medium and IFN-α at concentration of 100 U/mL), Cont. LPS—control of LPS (native HGF-1, medium and LPS at concentration of 200 ng/mL), Cont. LPS+IFN-α—control of LPS combined with IFN-α (native HGF-1, medium, LPS at concentration of 200 ng/mL, and IFN-α at concentration of 100 U/mL).

**Figure 5 pharmaceuticals-18-00628-f005:**
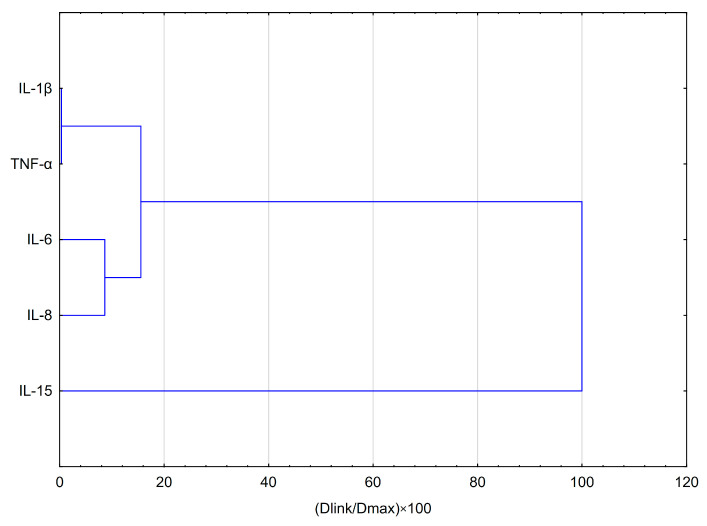
A dendrogram was obtained via the HCA analysis of all obtained data based on the average content of the effect of apigenin and quercetin at different concentrations on pro-inflammatory cytokines.

**Figure 6 pharmaceuticals-18-00628-f006:**
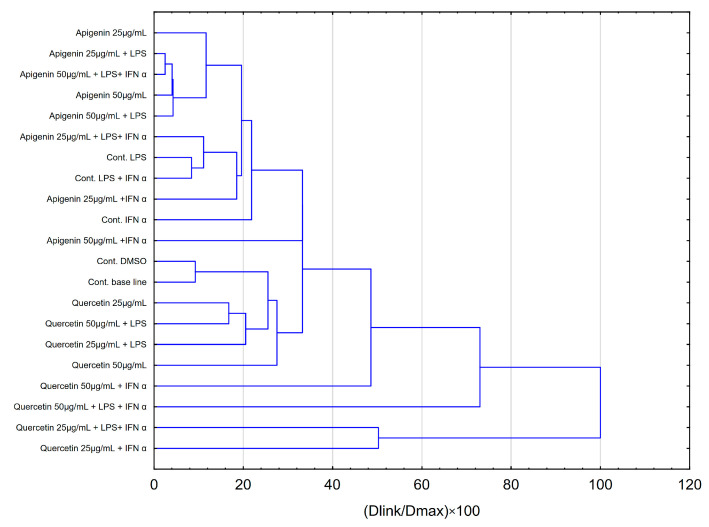
Dendrogram obtained via the HCA analysis of all obtained data based on the influence on pro-inflammatory cytokines by the average content of apigenin and quercetin at different concentrations.

**Figure 7 pharmaceuticals-18-00628-f007:**
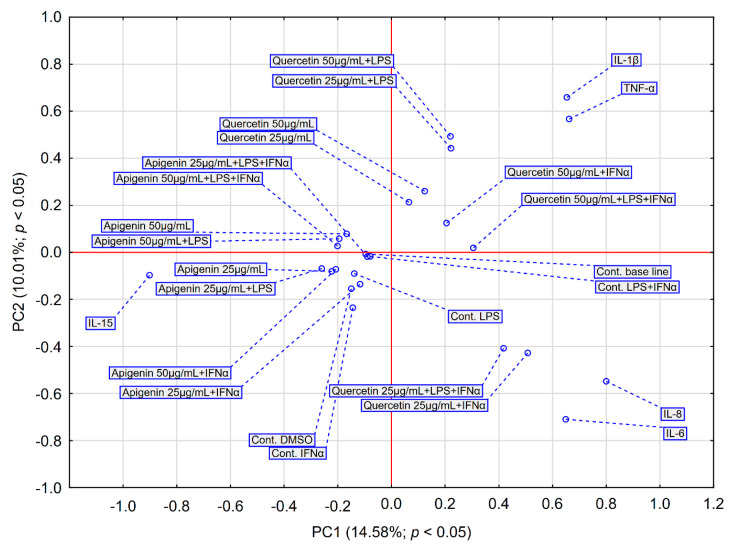
PCA score plot of all obtained data based on the average content of the effect of apigenin and quercetin at different concentrations on pro-inflammatory cytokines. PC—principal component.

## Data Availability

Data are contained within the article.

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
