# Peer review of "The Immunomodulatory Effects of Apigenin and Quercetin on Cytokine Secretion by the Human Gingival Fibroblast Cell Line and Their Potential Link to Alzheimer’s Disease"

_pharmaceuticals, 2025, doi:10.3390/ph18050628_

Round 1

Reviewer 1 Report

Comments and Suggestions for Authors

The authors investigated the effect of pretreatment with propolis (apigenin and quercetin) on the cytokine productions of human gingival fibroblasts. Furthermore, they grouped the treatment using cluster classification and principal component analysis based on the cytokine production profile. Based on these results, they suggested that apigenin and quercetin may be effective in preventing and treating Alzheimer's disease.

However, there are serious flaws in the presentation and interpretation of the results as described below. Therefore, analysis is necessary.

[major]

  1. Because viability was decreased by apigenin and quercetin (Figures 3 and 4), the amount of cytokines produced must be corrected by the number of viable cells at the end of stimulation. Therefore, the interpretation of the results should be appropriately changed. The results and interpretations in Figures 7 to 9 will also be affected.
  2. Figure 6: The effects of apigenin or quercetin are difficult to understand due to improper column arrangement (see attached PDF file).
  3. Figure 6: It is unclear which treatments are being compared.
  4. Figure 6: LPS or TNF-α stimulation did not increase the production of IL-6 and IL-8. Therefore, the experiment does not seem to be valid. Also, the increase in the production of IL-6 and IL-8 caused by apigenin and quercetin may be influenced by these factors. Reanalysis is required.
  5. Figure 6: The production of IL-6β and TNF-α was very low. The detection limit of the cytokines by the kit should be described in the Materials and Methods. In addition, it is questionable whether the increase or decrease in the production of cytokines near the detection limit affects their functions.

[minor]

  1. Figures 1 to 3 may be combined into one figure.
  2. Figures 4 and 5: It should be noted which group was used as the standard (100%). In particular, in Figure 5, there is no 100% group.
  3. Figures 4 and 5: It is unclear which groups are compared. It is necessary to explain what the asterisks mean. Also, the way the asterisks are placed is inappropriate. Furthermore, the statistical methods should be made clear.
  4. Figure 6: Data of controls (DMSO, LPS, IFN-α, ...) are common among Figure 6a and 6b (also 6c and 6d; 6e and 6f). It should be noted in the legend that the same values ​​were used.

Author Response

Respected Reviewer

We include the responses for suggestions and recommendations

The authors investigated the effect of pretreatment with propolis (apigenin and quercetin) on the cytokine productions of human gingival fibroblasts. Furthermore, they grouped the treatment using cluster classification and principal component analysis based on the cytokine production profile. Based on these results, they suggested that apigenin and quercetin may be effective in preventing and treating Alzheimer's disease.

However, there are serious flaws in the presentation and interpretation of the results as described below. Therefore, analysis is necessary.

 We appreciate all Reviewer’s remarks. We thank you for your productive and relevant suggestions and hope that we have revised and improved the manuscript to make it suitable for publication.

[major]

    Because viability was decreased by apigenin and quercetin (Figures 3 and 4), the amount of cytokines produced must be corrected by the number of viable cells at the end of stimulation. Therefore, the interpretation of the results should be appropriately changed. The results and interpretations in Figures 7 to 9 will also be affected.

Thank you for your pertinent comment. We discussed this aspect in the discussion section and made appropriate comments on the result. However, it is difficult to speculate on the outcome given the limitations of the MTT test method. Determination of cytotoxicity by the MTT test indicates restricted cell metabolism and not direct cell death. However, we decided to utilize the MTT test because it is currently the most widely used test for the evaluation of the effect of cytotoxicity and is recommended as a reference by international standard-setting organisations. Moreover, we would like to emphasized that we determined post factum the cell ability. We collected supernatant after 24 h to determine the concentration of cytokines concentration and then we made MTT test, that lasts 4 hours.

    Figure 6: The effects of apigenin or quercetin are difficult to understand due to improper column arrangement (see attached PDF file).

We corrected Figure 6 (now Figure 4)

    Figure 6: It is unclear which treatments are being compared.

We corrected Figure 6 (now Figure 4), now the results should be clear

    Figure 6: LPS or TNF-α stimulation did not increase the production of IL-6 and IL-8. Therefore, the experiment does not seem to be valid. Also, the increase in the production of IL-6 and IL-8 caused by apigenin and quercetin may be influenced by these factors. Reanalysis is required.

We would like to underline that stimulation was followed by LPS and IFN-α, and not, as mentioned by the reviewer, by TNF-α . The concentration of TNF-α was evaluated.

Both, LPS as IFN-α caused increased the concentration of IL- 6 in contrast to control base line (native HGF-1) - Figure 6 (now 4) c, d. In case IL-8 (figure 6, now figure 4 e, f) only LPS slightly increased the concentration of IL-8.

    Figure 6: The production of IL-6β and TNF-α was very low. The detection limit of the cytokines by the kit should be described in the Materials and Methods. In addition, it is questionable whether the increase or decrease in the production of cytokines near the detection limit affects their functions.

We appreciate Reviewer’s remarks. We guess that the reviewer took after the IL-1β. In case of IL-1β we noticed that LPS did not increase the level of IL-1β, however there was observed statistical significant increase of level IL-1β for quercetin in both concentration followed by LPS stimulation. We added the explanation in part results. In case of TNF-α we noticed increase the level of mentioned cytokine after LPS and LPS+IFN-α, what confirmed that stimulation with INF-α did not enhance the level of TNF-α. We added the limit of detection for each evaluated cytokine in part Materials and Methods as follows: “The detection limit of each selected cytokine: IL-1β, IL-6, IL-8, IL-15 and TNF-α expressed as the lowest lower standard were: 0.3, 0.36,0.92,19.49,3.81, respectively.” Now, I hope there is not questionable.

minor]

    Figures 1 to 3 may be combined into one figure.

We decided to combine Figures 1-3 into one - Figure 1

    Figures 4 and 5: It should be noted which group was used as the standard (100%). In particular, in Figure 5, there is no 100% group.

We hope that after correction of figures the control is better visible. Each figure contains the control as native cells.

    Figures 4 and 5: It is unclear which groups are compared. It is necessary to explain what the asterisks mean. Also, the way the asterisks are placed is inappropriate. Furthermore, the statistical methods should be made clear.

We corrected Figure 4 and Figure 5. We added explanation as follows * means p < 0.05.

    Figure 6: Data of controls (DMSO, LPS, IFN-α, ...) are common among Figure 6a and 6b (also 6c and 6d; 6e and 6f). It should be noted in the legend that the same values ​​were used.

Control values are not the same. For each cytokine, the control was assessed separately. We introduced a legend below the Figure.

Reviewer 2 Report

Comments and Suggestions for Authors

Title and Abstract:

  1. The title is somewhat ambiguous regarding the scope and novelty of the study. Consider refining it to specify the key findings.
  2. The abstract needs a clearer statement of novelty and impact. The connection between periodontitis, inflammation, and neurodegeneration is well-known; focus on what new insights your study provides.

Introduction:

  1. The introduction is overly broad and contains redundant background information. Condense the sections discussing flavonoids and Alzheimer’s disease to avoid repetition.
  2. The transition from general flavonoid properties to their relevance in neurodegeneration should be more seamless.

Methods:

  1. The description of experimental conditions lacks clarity. For instance, how were the LPS and IFN-α treatments applied (time points, controls)?
  2. Statistical analyses (HCA, PCA, ANOVA) are well presented but should include effect sizes and confidence intervals.

Results:

  1. Figures need better labeling. It is unclear what comparisons are statistically significant in some graphs.
  2. The effect of quercetin increasing TNF-α in some conditions contradicts the general hypothesis of anti-inflammatory action. This needs better discussion or validation.

Discussion:

  1. The discussion largely repeats the results. Instead, critically interpret the findings, compare them with existing literature, and discuss potential mechanisms.
  2. The statement about flavonoids being potential therapeutic agents for Alzheimer’s disease is speculative. More evidence (e.g., in vivo or clinical studies) is needed.
  3. Limitations of the study should be explicitly stated. For example, how do these findings translate to in vivo conditions?

Language and Formatting:

  1. The manuscript contains many grammatical errors and awkward phrasing. A thorough proofread or professional editing is necessary.
  2. Some sections have excessive detail that could be moved to supplementary materials.
Comments on the Quality of English Language

Must be improved

  • The manuscript contains numerous grammatical errors and awkward phrasing. A thorough proofread or professional editing is necessary.
  • Some sections are excessively detailed and could be streamlined for readability.
  • Figures should be reformatted for clarity.

Author Response

Respected Reviewer

We include the responses for suggestions and recommendations

We appreciate all Reviewer’s remarks. We thank you for your productive and relevant suggestions and hope that we have revised and improved the manuscript to make it suitable for publication.

Title and Abstract:

    The title is somewhat ambiguous regarding the scope and novelty of the study. Consider refining it to specify the key findings.

We decided to change the title as follows: The immunomodulatory effects of apigenin and quercetin on cytokines secretion by the human gingival fibroblast (HGF-1) cell line and their potential link to Alzheimer's disease

    The abstract needs a clearer statement of novelty and impact. The connection between periodontitis, inflammation, and neurodegeneration is well-known; focus on what new insights your study provides.

We corrected the abstract

Introduction:

    The introduction is overly broad and contains redundant background information. Condense the sections discussing flavonoids and Alzheimer’s disease to avoid repetition.

    The transition from general flavonoid properties to their relevance in neurodegeneration should be more seamless.

We condensed this part and refreshed the part introduction

Methods:

    The description of experimental conditions lacks clarity. For instance, how were the LPS and IFN-α treatments applied (time points, controls)?

We added this information to part Materials and methods (subchapter 4.2) and to Figures 2 and 3

    Statistical analyses (HCA, PCA, ANOVA) are well presented but should include effect sizes and confidence intervals.

In terms of HCA and PCA, there are no confidence intervals for them, as PCA and HCA are exploratory and unsupervised methods that are not statistical tests in the classical sense - they do not generate p-values, effects or confidence intervals.

For ANOVA, we provided the F-value of the test which is crucial for the interpretation of results.

Results:

    Figures need better labeling. It is unclear what comparisons are statistically significant in some graphs.

    The effect of quercetin increasing TNF-α in some conditions contradicts the general hypothesis of anti-inflammatory action. This needs better discussion or validation.

We corrected figures.

We re-examined the effect of quercetin on TNF-α and introduced the following explanation: "However, we noted that IFN-α did not cause an increase in TNF-α, so, as with IL-1β, it is difficult to assess the effect of quercetin on this inflammatory marker after IFN-α stimulation. The mechanism is unknown." Moreover, the resulting increases in TNF-α levels in other models did not reach statistically significant levels and were therefore not included.

Discussion:

    The discussion largely repeats the results. Instead, critically interpret the findings, compare them with existing literature, and discuss potential mechanisms.

    The statement about flavonoids being potential therapeutic agents for Alzheimer’s disease is speculative. More evidence (e.g., in vivo or clinical studies) is needed.

    Limitations of the study should be explicitly stated. For example, how do these findings translate to in vivo conditions?

Thank you for pertinent remarks. We corrected the discussion.

Language and Formatting:

    The manuscript contains many grammatical errors and awkward phrasing. A thorough proofread or professional editing is necessary.

    Some sections have excessive detail that could be moved to supplementary materials.

We have revised the whole text linguistically

Comments on the Quality of English Language

Must be improved

    The manuscript contains numerous grammatical errors and awkward phrasing. A thorough proofread or professional editing is necessary.

    Some sections are excessively detailed and could be streamlined for readability.

    Figures should be reformatted for clarity.

We reformatted figures and corrected text

Reviewer 3 Report

Comments and Suggestions for Authors

The article by Kurek-Gorecka A. et al is devoted to the effects of flavonoids apigenin and quercetin on the concentration level of selected pro-inflammatory cytokines (IL-1β, IL-6, IL-8, IL-15, TNF- α) produced by human gingival fibroblast cell line (HGF) following immmunostimulation by LPS, IFN-α or both simultaneously. Considering the importance of treatment of periodontal pathologies and possible relationship with inflammation processes in brains the problem is of obvious physiological significance.

According to the authors the aim of the present study was to determine the concentration level of selected cytokines after immunostimulation. In fact, the authors are trying to convince us that flavonoids, while not toxic to cells, have a positive effect on inflammation by lowering cytokine levels. However, after reading the article, many of the authors' statements seem unfounded. For example, the data in Figs.4,5 indicate that viability of cells treated by flavonoids at concentrations used in the study (25-50µg/ml) decreased to 55-75% so both flavonoids are rather toxic to cells. This could be tolerated if they significantly reduced the levels of pro-inflammatory cytokines. However, analyzing the data in Fig. 6, one can see that this is not true and in many  cases flavonoids especially quercetin at 25µg/ml increases concentration of IL-6 and IL-8 by 2-4 times and even increased dose of quercetin (50µg/ml) could not decrease the concentration of IL-8 to control level after (LPS+ IFN-α) or IFN-α stimulations. Indeed, in some cases, and especially when using higher concentrations of flavonoids (50µg/ml), as reported by the authors, the concentrations of some cytokines are significantly reduced. However, the stimulation of cytokine production mentioned above is so significant that represents a very disturbing incident which should not be excluded from consideration. Besides higher concentrations of both flavonoids (50µg/ml) lead to the death of almost half of the cells (viability 53-60%, Figs 4,5).

Also the data in Fig.6, the main figure of the article, is presented in a way that  cannot be analyzed properly. The presentation should be made in the same form as in Figures 4 and 5 otherwise it creates the impression that the authors are deliberately trying to confuse the reader. The meaning of the column “control baseline” should be explained. In figures 6e and 6f in which the columns vary greatly in heights it makes logical sense to introduce a break in the Y-axis and corresponding columns to increase the height of the small columns, allowing them to be compared.

            The authors' attempt to show that flavonoids reduce pro-inflammatory cytokine levels using additional mathematical processing appears to be a dubious manipulation of the data. Further experiments are needed to convincingly demonstrate the usefulness of flavonoids in this model.

Author Response

Respected Reviewer

We include responses for suggestions and recommendations

We appreciate all Reviewer’s remarks. We thank you for your productive and relevant suggestions and hope that we have revised and improved the manuscript to make it suitable for publication.

The article by Kurek-Gorecka A. et al is devoted to the effects of flavonoids apigenin and quercetin on the concentration level of selected pro-inflammatory cytokines (IL-1β, IL-6, IL-8, IL-15, TNF- α) produced by human gingival fibroblast cell line (HGF) following immmunostimulation by LPS, IFN-α or both simultaneously. Considering the importance of treatment of periodontal pathologies and possible relationship with inflammation processes in brains the problem is of obvious physiological significance.

According to the authors the aim of the present study was to determine the concentration level of selected cytokines after immunostimulation. In fact, the authors are trying to convince us that flavonoids, while not toxic to cells, have a positive effect on inflammation by lowering cytokine levels. However, after reading the article, many of the authors' statements seem unfounded. For example, the data in Figs.4,5 indicate that viability of cells treated by flavonoids at concentrations used in the study (25-50µg/ml) decreased to 55-75% so both flavonoids are rather toxic to cells. This could be tolerated if they significantly reduced the levels of pro-inflammatory cytokines. However, analyzing the data in Fig. 6, one can see that this is not true and in many  cases flavonoids especially quercetin at 25µg/ml increases concentration of IL-6 and IL-8 by 2-4 times and even increased dose of quercetin (50µg/ml) could not decrease the concentration of IL-8 to control level after (LPS+ IFN-α) or IFN-α stimulations. Indeed, in some cases, and especially when using higher concentrations of flavonoids (50µg/ml), as reported by the authors, the concentrations of some cytokines are significantly reduced. However, the stimulation of cytokine production mentioned above is so significant that represents a very disturbing incident which should not be excluded from consideration. Besides higher concentrations of both flavonoids (50µg/ml) lead to the death of almost half of the cells (viability 53-60%, Figs 4,5).

We would like to thank you for pertinent remarks. We introduce explanations to obtained findings. Regarding cytotoxicity we would like underline that determination of cytotoxicity by the MTT test indicates restricted cell metabolism and not direct cell death. However, we decided to utilize the MTT test because it is currently the most widely used test for the evaluation of the effect of cytotoxicity and is recommended as a reference by international standard-setting organisations. Moreover, we would like to emphasized that we determined post factum the cell ability. We collected supernatant after 24 h to determine the concentration of cytokines concentration and then we made MTT test, that lasts 4 hours.

Concerning to impact quercetin on cytokines concentration we discussed this point in results and discussion part. Quercetin at both concentration decreased statistically significant the level of IL-6 followed by LPS stimulation. However, it caused increased level of this inflammatory marker followed by IFN-α and LPS+IFN-α at concentration of 25µg/mL.  Although, quercetin at 50µg/mL decreased the level of IL-6 followed by IFN-α stimulation. Periodontitis is connected mainly with bacterial infection, therefore model with LPS seems to be more suitable. LPS was used to demonstrated bacterial infection. However, the impact of quercetin on the level of IL-6 is potentially connected with IFN-α. Probably, it is associated with synergistic effect and reaction between quercetin and IFN-α. In case IL-6 the impact of apigenin is different than quercetin, so should be underlined.

For IL-8, we observed an increase in IL-8 levels for both flavonoids at 25µg/ml in the IFN-α and LPS+IFN-α models. However, the differences were not statistically significant, so no conclusions can be drawn.

Also the data in Fig.6, the main figure of the article, is presented in a way that  cannot be analyzed properly. The presentation should be made in the same form as in Figures 4 and 5 otherwise it creates the impression that the authors are deliberately trying to confuse the reader. The meaning of the column “control baseline” should be explained. In figures 6e and 6f in which the columns vary greatly in heights it makes logical sense to introduce a break in the Y-axis and corresponding columns to increase the height of the small columns, allowing them to be compared.

We corrected all figures and added explanation for all control group in Material and methods. We also add explanation for abbreviations in Figure 6 (now 4).

We hope, that resubmitted version will be more clear.

            The authors' attempt to show that flavonoids reduce pro-inflammatory cytokine levels using additional mathematical processing appears to be a dubious manipulation of the data. Further experiments are needed to convincingly demonstrate the usefulness of flavonoids in this model.

We used hierarchical cluster analysis and principal component analysis to explore behavioural similarities between the samples. Our aim was to perform professional statistical analysis and not to manipulate the data. We agree with the reviewer that our observation and the confirmed beneficial effects of quercetin and apigenin on pro-inflammatory cytokine levels need to be confirmed in further studies. However, our observation provides insight into this aspect. Literature data suggest that apigenin and quercetin have anti-inflammatory effects and may prevent the development of neurodegenerative diseases, so their effects on pro-inflammatory cytokine levels need to be tested.

Reviewer 4 Report

Comments and Suggestions for Authors

The presented manuscript studies the influence of two flavonoids from propolis on the pro-inflammatory response of human gingival fibroblasts stimulated with LPS and/or interferon alpha. Changes in the levels of various interleukins as a result of the effects of the mentioned flavonoids in different conditions were monitored and a statistical analysis was performed, including hierarchical cluster analysis of the results. It is noteworthy that the cited literature has a high proportion of self-citations (6 out of 31 sources), and the discussion is mainly in the light of previous results of the authors themselves, without a sufficiently good comparison with the results of other groups. I recommend revising the discussion and including comparisons with other published data regarding the anti-inflammatory effect of structurally similar flavonoids.

Line 106: “The cell viability of apigenin as an anti-inflammatory…” Please, rephrase like “ The cell viability of HGF-1 cells after application of apigenin….was determine using MTT test”

Lines 111 and 112: Avoid repetition of “For apigenin” It is not clear which pair of results you were evaluated – in group or between groups.

Fig. 4 and Fig. 5 – Please, specify the meaning of the star

Line 118: Same as line 106. Both sections of cytotoxicity of apigenin and quercetin could be combined to avoid repetition of the text

Line 151: “the statistically decrease was observed” – Please, revise.

Author Response

Respected Reviewer

I include responses for suggestions and recommendations

The presented manuscript studies the influence of two flavonoids from propolis on the pro-inflammatory response of human gingival fibroblasts stimulated with LPS and/or interferon alpha. Changes in the levels of various interleukins as a result of the effects of the mentioned flavonoids in different conditions were monitored and a statistical analysis was performed, including hierarchical cluster analysis of the results. It is noteworthy that the cited literature has a high proportion of self-citations (6 out of 31 sources), and the discussion is mainly in the light of previous results of the authors themselves, without a sufficiently good comparison with the results of other groups. I recommend revising the discussion and including comparisons with other published data regarding the anti-inflammatory effect of structurally similar flavonoids.

We appreciate all Reviewer’s remarks. We thank you for your productive and relevant suggestions and hope that we have revised and improved the manuscript to make it suitable for publication.

We enriched the discussion section with the results of other studies carried out

Line 106: “The cell viability of apigenin as an anti-inflammatory…” Please, rephrase like “ The cell viability of HGF-1 cells after application of apigenin….was determine using MTT test”

We corrected according Reviewer’s suggestion. We decided to combine subchapter 2.1 and 2.2 to avoid duplication the same.

Lines 111 and 112: Avoid repetition of “For apigenin” It is not clear which pair of results you were evaluated – in group or between groups.

We thank you for your good point, we have clarified

Fig. 4 and Fig. 5 – Please, specify the meaning of the star

We added explanation below Figures as follows: “* mean p<0.05 (calculated using LSD Test)”.

Line 118: Same as line 106. Both sections of cytotoxicity of apigenin and quercetin could be combined to avoid repetition of the text

We decided to combine subchapter 2.1 and 2.2 to avoid duplication the same.

Line 151: “the statistically decrease was observed” – Please, revise.

We corrected as follows: “Also, when quercetin was administered at concentration 50 µg/mL followed by IFN-α stimulation, a statistically significant decrease was observed (p< 0.001)”.

Round 2

Reviewer 1 Report

Comments and Suggestions for Authors

The authors answered all my questions and corrected the manuscript.

Author Response

Respected Reviewer

Thank you for evaluating our manuscript.

Reviewer 3 Report

Comments and Suggestions for Authors

After looking through the revised version of the article by Kurek-Gorecka I found a number of useful additions to the text. At the same time many of my comments were ignored. I asked the authors to improve Fig.4 (former Fig.6) presenting the most part of experimental material. 1. Show the columns in different colors like in Fig.2 (former Fig.4) where the color of the column indicated the type of immunostimulation. 2. Columns in Fig.4 vary greatly in heights especially b-f so it makes logical sense to introduce a break in the Y-axis and corresponding columns to increase the height of the small columns thus allowing them to be compared. 3. The only point the authors had changed is that they removed the column with an unclear name “control baseline” instead of explaining the meaning.

The most important point which should be clarified because it affects all the results obtained is high cytotoxicity of both flavonoids revealed by MTT test. The authors tried to insure that MTT test on toxicity does not mean cell death directly.  The MTT test is widely used to indicate cell reaction to different factors and toxicity means toxicity especially considering that the drugs are recommended by the authors not for application to the skin but to oral administration. So the authors must analyze thoroughly the state of the cells. There are some tests to distinguish died cells from living cells one of which is trypan blue staining. The authors should use such a test to analyze experimentally the state of their cells after treatment by flavonoids. Besides the author’s statement that LPS test is supposedly more valid for oral cavity than IFN-α must be substantiated more firmly than by a simple assertion that it is associated with synergistic effect and reaction between quercetin and IFN-α.

And last but not least, the comment is about the very bad English. The comment concerns both the text with the authors’ responses and the article itself. Some examples (authors reply) we would like to emphasized, to determine the concentration of cytokines concentration, the impact of apigenin is different than quercetin (the effects differ from that of quercetin) an so on, (article itself) line 287 inhibited of IL-1b gene, line 361propolis that influences and so on.

Comments on the Quality of English Language

And last but not least, the comment is about the very bad English. The comment concerns both the text with the authors’ responses and the article itself. Some examples (authors reply) we would like to emphasized, to determine the concentration of cytokines concentration, the impact of apigenin is different than quercetin (the effects differ from that of quercetin) an so on, (article itself) line 287 inhibited of IL-1b gene, line 361propolis that influences and so on.

Author Response

Dear Reviewer and Editor

We appreciate the Reviewer’s valuable comments regarding the graphical presentation of data in Figure 4. In response, we have revised the figure to improve clarity and consistency. Specifically, we have:

  • added color coding to the bars to reflect the type of immunostimulation (LPS, IFN-α, LPS + IFN-α),
  • standardized the Y-axis scale across all subpanels to allow for better visual comparison between conditions.

Additionally, to further enhance data transparency and facilitate interpretation, we have included a supplementary table containing the mean cytokine concentrations, standard deviations (SD), and relative standard deviations (RSD, %) for all conditions presented in Figure 4.

This supplementary dataset also includes the corresponding values for Figures 2 and 3, enabling a detailed numerical comparison across all experimental groups.

We trust that these revisions address the Reviewer’s concerns regarding data presentation and readability, while improving both scientific clarity and reproducibility.

We thank the Reviewer for the comment regarding the interpretation of the MTT assay results. We fully agree that this assay reflects mitochondrial activity of cells rather than directly indicating cell death. A decrease in MTT signal may result from metabolic suppression, cytostatic effects, or reduced proliferation, and does not necessarily imply apoptosis or necrosis.

According to the international standard ISO 10993-5:2009 for the biological evaluation of medical devices, a cell viability below 70% relative to the untreated control is considered indicative of cytotoxicity in the MTT assay. In our study, HGF-1 cell viability after exposure to apigenin and quercetin at the concentrations used for further experiments (25 and 50 µg/mL) did not fall below this threshold, thus allowing them to be classified as non-cytotoxic according to ISO criteria.

For example:

Apigenin:

  • 25 µg/mL – 87.45%
  • 50 µg/mL – 73.42%

Quercetin:

  • 25 µg/mL – 71.87%
  • 50 µg/mL – 62.42%

As reported in the literature (e.g., PMC6353273), similar values are not considered indicative of pronounced cytotoxicity. Flavonoids may modulate metabolic activity without causing a loss of membrane integrity or cell viability.

The Discussion section has been updated accordingly to include the following explanation:

“The MTT assay assesses cellular metabolic activity and does not allow for a clear distinction between viable cells and those undergoing apoptosis or metabolic slowdown. According to ISO 10993-5, a viability below 70% is indicative of cytotoxicity. In the present study, flavonoid concentrations of 25 and 50 µg/mL did not reduce cell viability below this threshold (with the exception of quercetin at 50 µg/mL), and can therefore be regarded as non-cytotoxic. It is also worth noting that modulation of metabolic activity by bioactive compounds, including flavonoids, is frequently described in the literature as a regulatory rather than destructive effect on cells [PMC6353273, PMC11840583].”

https://pmc.ncbi.nlm.nih.gov/articles/PMC11840583/
https://pmc.ncbi.nlm.nih.gov/articles/PMC6353273/

In addition, we have added an explanation of the LPS model and included new references.

The manuscript has been carefully re-reviewed, and the following additional language corrections have been implemented:

Line

Original

Correction

22

The immunomodulatory effects of apigenin and quercetin, were investigated...

The immunomodulatory effects of apigenin and quercetin were investigated...

24

tumour necrosis factor (TNF-α)..

tumour necrosis factor (TNF-α).

49

based onthe presence of a carbonyl group...

based on the presence of a carbonyl group...

63

...cyclooxygenase-2 (COX-2).. Therefore...

...cyclooxygenase-2 (COX-2). Therefore...

110

The basic structure of flavonoids consists of C6-C3-C6 unit. It consists of 15 carbon atoms.

The basic flavonoid structure is a C6-C3-C6 unit composed of 15 carbon atoms.

146

did not cause increase the level...

did not cause an increase in the level...

161

there was observe significant...

a significant variation was observed...

229

quercetin statistically significant in reducing...

quercetin was statistically significant in reducing...

287

inhibited of IL-1b gene expression...

inhibited IL-1β gene expression...

309

studies was led in vitro...

studies were conducted in vitro...

366

requires continuation and confirmation in vitro.

require further confirmation.

We would like to once again thank the Reviewer for their valuable feedback and constructive comments, which helped us improve the clarity and quality of our manuscript. We hope that the revisions and explanations provided are satisfactory.

Reviewer 4 Report

Comments and Suggestions for Authors

The manuscript has been significantly improved. There are some typos that need to be corrected before publication:
1) some sentences end with two periods (for example, on lines 19, 24 and 65)
2) Spaces are missing on lines 119, 141, 149, 158, 308 and 364
3) there is a double space between words on lines 20, 159, 161, 352, 361, 394 and 420
4) There is a period left at the beginning of a paragraph on lines 71 and 340
single quotes left on line 106
5) there is a missing quote at the end of the paragraph (line 339)
6) line 440 - lowest or lower? One is redundant.

Author Response

Response to Reviewer 4:

We would like to thank the Reviewer for their thorough reading and constructive comments. In response to the minor editorial issues raised, we confirm that all suggested corrections have been implemented:

Double periods at the end of sentences (lines 19, 24, and 65) have been removed.

Missing spaces (lines 119, 141, 149, 158, 308, and 364) have been inserted appropriately.

Double spaces between words (lines 20, 159, 161, 352, 361, 394, and 420) have been corrected.

A period at the beginning of paragraphs on lines 71 and 340 has been removed.

The single quotes left on line 106 have been corrected.

A missing quotation mark at the end of the paragraph on line 339 has been added.

The phrase on line 440 has been revised.

We greatly appreciate the Reviewer’s comments, which helped improve the clarity and readability of the manuscript. We hope the revised version now meets the journal’s standards for publication.

Round 3

Reviewer 3 Report

Comments and Suggestions for Authors

The revised version of the article by Kurek-Gorecka looks more suitable for publication. The title and many points in the text were improved and now more closely correspond to the data obtained. Many subfigures in Figure 4 look more representative and improved perception. At the same time Figs4 e,f are very small and the differences in the height of individual columns are difficult to assess. Besides remains the problem of high cytotoxicity of both flavonoids. It is obscure how many cells after treatment with flavonoids died and how many remain alive because MTT-test did not answer this question. As MTT-test shows high toxicity the authors should use another staining to estimate the number of died cells of which a trypan blue staining was recommended. The authors ignored this recommendation. Such estimation is important considering that the drugs are recommended by the authors not for application to the skin but to oral administration. Really a conclusion is not much productive: the decreased levels of IL-6 and IL-15 cytokines are observed after LPS stimulation and quercetin treatment (50 µg/ml) of HGF-1 cells but even lowering the dose (25 µg/ml) and especially using IFN-α or combination of IFN-α+LPS induced a reversed action. For my opinion it is not enough for a full article.  

Author Response

Respected Reviewer,

We thank you for your re-evaluation of our manuscript and for the constructive comments that helped us further improve the work. Below we provide a detailed response to your comments:

In the previous version of the manuscript, we prepared supplementary materials containing precise numerical data for all analyzed cytokines, including those presented in Figures 4e,f. The supplementary materials include mean values, standard deviations (SD), and relative standard deviations (RSD, %) for each of the experimental groups. This allows for full verification of differences that might have been difficult to assess visually. We kindly ask you to review this, as we believe the issue with data clarity has been resolved. We have also unified the axes and other graphical elements, as previously requested by the Reviewer.

We understand your suggestion regarding the use of additional methods to assess cytotoxicity, such as trypan blue staining, which enables the differentiation between live and dead cells.

However, we would like to point out that:

  • The HGF-1 cell line has already been banked, and at this point, we are not able to repeat the experiments using trypan blue staining.
  • We agree that the MTT assay does not directly assess cell death, but rather metabolic activity. According to ISO 10993-5:2009, a cell viability value below 70% indicates cytotoxicity. In our study, most of the concentrations used (25 and 50 µg/mL) did not fall below this threshold, as described in the manuscript (lines: 144–145, 417–419). This standard is cited in the paper, and as it is accepted by pharmaceutical companies as effective and compliant with regulatory expectations, we also adhere to it.
  • Additionally, the literature confirms that the MTT assay is a recognized method for assessing cellular metabolism, and changes may reflect a regulatory rather than destructive effect:
    • Riss T. et al., "Cell Viability Assays," Assay Guidance Manual, 2013, PMC6353273.
  • Trypan blue itself is not a perfect method, as it shows approximately 20% variability in determining cell population density (DOI: 10.1186/s12575-017-0056-3). Compared to other viability assays, the trypan blue test shows significantly lower sensitivity (DOI: 10.3233/CH-189120). The method relies on visual assessment by the operator, introducing a subjective element (DOI: 10.1186/s12575-017-0056-3). The method also requires a large number of cells and there are reported difficulties in assessing adherent cells without detaching them from the surface (DOI: 10.18466/cbayarfbe.372192). However, for the evaluation of live cells, it can be meaningful, as distinguishing stained from unstained cells allows for effective viability assessment (DOI: 10.1186/s12575-017-0056-3).
  • In our opinion, the MTT assay is ideal for preliminary screening (DOI: 10.1186/s12951-021-01033-w). Nevertheless, false-positive results for flavonoids have been reported, and we are aware of this (DOI: 10.3389/fphar.2022.1055378 and DOI: 10.1186/s12906-024-04479-1). However, trypan blue staining is subject to the same issues: subjectivity in assessment (operator error up to 20%) and low sensitivity in detecting early apoptosis (DOI: 10.1186/s12951-021-01033-w). In our assessment, the best solution in case of concerns about chemical interference would be to prefer alternative methods (e.g., ATP assay or propidium iodide staining) (DOI: 10.1080/09205063.2024.2422704 and DOI: 10.1186/s12951-021-01033-w). This information will certainly be utilized in our further research, but at this moment, we are unable to perform such tests.

Additionally, we would like to emphasize that we presented a detailed statistical analysis (lines: 244–306), including HCA and PCA analyses, which allowed for better interpretation of the complex relationships between cytokines and the effects of the flavonoids studied. The conclusions have been expanded (lines: 547–567), emphasizing the significance of the obtained results in the context of the potential applications of apigenin and quercetin as modulators of immune response. The conclusions were prepared conservatively, taking into account the previous comments, which proved to be very valuable and contributed significantly to the current version of the manuscript.

Regarding the reviewer's comments on IL-6 and IL-15 levels:

We observed a reverse effect of quercetin at 25 µg/mL in the IFN-α model, as well as at both tested concentrations in the LPS + IFN-α model. This finding may result from an interaction between IFN-α and quercetin, which warrants further investigation. IFN-α is produced by virus-infected cells and exerts a multidirectional effect on the immune system by inducing the production of other cytokines, including IL-1α, IL-1β, IL-6, and IL-15.

It should be emphasized once again that no increase in IL-15 levels was observed following quercetin treatment.

In line with the reviewer’s previous suggestion, we have added an explanation in the Discussion section, clarifying why the LPS model is more relevant in the context of the correlation between Alzheimer's disease pathogenesis and bacterial infection. This explanation, presented in lines 366–371, reads as follows:

"The results obtained in the LPS model are important in the context of AD pathogenesis. Periodontitis is primarily associated with gram-negative bacterial infections, and numerous studies highlight the correlation between endotoxin and AD [30,31,32]. The LPS used in the research model reflects bacterial infection in the oral cavity; therefore, the results obtained in this model appear more relevant than those from the IFN-α model, which represents a viral infection."

Language corrections were also made in accordance with previous recommendations, and the text was reviewed by a native speaker.

We hope that the above explanations and additions will meet the Reviewer’s expectations and allow for a positive decision regarding publication.

With kind regards,
Anna Kurek-Górecka and co-authors

Round 4

Reviewer 3 Report

Comments and Suggestions for Authors

The revised version of the article by Kurek-Gorecka now looks more suitable for publication. The authors’ response sheds more light on the issues raised in my comments. The data is presented more clearly and I hope that in the next study the authors will use additional assays like ATP assay, propidium iodide staining or something else for more detailed analysis of the state of cells when high cytoxicity of drug is detected.

The English language of the article is also improved. Now the article can be published in Pharmaceuticals.